# Improving atomic displacement and replacement calculations with physically realistic damage models

Kai Nordlund [1], Steven J. Zinkle [2,3], Andrea E. Sand [1], Fredric Granberg [1], Robert S. Averback[4], Roger Stoller[3], Tomoaki Suzudo [5], Lorenzo Malerba[6], Florian Banhart[7], William J. Weber [3,8], Francois Willaime [9], Sergei L. Dudarev [10] & David Simeone[11]

Atomic collision processes are fundamental to numerous advanced materials technologies such as electron microscopy, semiconductor processing and nuclear power generation. Extensive experimental and computer simulation studies over the past several decades provide the physical basis for understanding the atomic-scale processes occurring during primary displacement events. The current international standard for quantifying this energetic particle damage, the Norgett—Robinson—Torrens displacements per atom (NRT-dpa) model, has nowadays several well-known limitations. In particular, the number of radiation defects produced in energetic cascades in metals is only ~1/3 the NRT-dpa prediction, while the number of atoms involved in atomic mixing is about a factor of 30 larger than the dpa value. Here we propose two new complementary displacement production estimators (athermal recombination corrected dpa, arc-dpa) and atomic mixing (replacements per atom, rpa) functions that extend the NRT-dpa by providing more physically realistic descriptions of primary defect creation in materials and may become additional standard measures for radiation damage quantification.

[1] Department of Physics, University of Helsinki, P.O.Box 43, Helsinki FI-00014, Finland. [2] Department of Nuclear Engineering, University of Tennessee, Knoxville, TN 37996, USA. [3] Materials Science & Technology Division, Oak Ridge National Laboratory, P.O. Box 2008, Oak Ridge, TN 37831, USA. [4] Department of Materials Science & Engineering, University of Illinois, Urbana, IL 61801, USA. [5] Japan Atomic Energy Agency Center for Computational Science and e-Systems, Tokai, Ibaraki 319-1195 Japan. [6] SCK-CEN, Institute for Nuclear Materials Science, 2400 Mol, Belgium. [7] Institut de Physique et Chimie des Matériaux, CNRS, UMR 7504, Université de Strasbourg, 67000 Strasbourg, France. [8] Department of Materials Science and Engineering, University of Tennessee, Knoxville, TN 37996, USA. [9] DEN-Département des Matériaux pour le Nucléaire, CEA, Université Paris-Saclay, 91191 Gif-sur-Yvette, France. [10] Culham Centre for Fusion Energy, UK Atomic Energy Authority, Abingdon, Oxfordshire OX14 3DB, UK. [11] DEN/DMN/SRMA/LA2M-LRC CARMEN, CEA, Université Paris-Saclay, 91191 Gif-sur-Yvette, France. Correspondence and requests for materials should be addressed to K.N. (email: kai.nordlund@helsinki.fi)

Quantification of the amount of displacement damage introduced by energetic particle interactions in matter is important for a broad range of fundamental science and applied engineering applications ranging from semiconductor physics to nuclear energy generation[1]. Kinchin and Pease[2] developed the basis for an early model to calculate displacements per atom (dpa) by considering kinetic energy transfers above a threshold material-specific displacement energy. The current de facto international standard for quantifying atomic displacement levels in irradiated materials is based on the more than 40-year-old binary collision computer simulations of ion collisions in solids[3,4]. The predicted number of atomic displacements ($N_d$) as a function of cascade energy, or the damage function, is given in this model by

$$N_d(T_d) = \begin{bmatrix} 0 & , & T_d < E_d \\ 1 & , & E_d < T_d < \frac{2E_d}{0.8} \\ \frac{0.8 T_d}{2 E_d} & , & \frac{2 E_d}{0.8} < T_d < \infty \end{bmatrix}, \quad (1)$$

where $T_d$ is the damage energy, i.e. the kinetic energy available for creating atomic displacements. The damage energy for a single ion is given by the total ion energy minus the energy lost to electronic interactions (ionization). Typical values of $E_d$ for different materials range from 20 to 100 eV[5,6]. This is essentially the Kinchin–Pease model, except that the original kinetic energy term was replaced by the damage energy to account for ionization effects and a factor of 0.8 was introduced to account for more realistic interatomic potentials.

The importance of the calculated dpa parameter is that it is the starting point for calculations of virtually all radiation effects in solid materials, and it facilitates quantitative comparisons of different materials irradiated with the same irradiation source as well as materials irradiated in different irradiation sources such as electron, ion and neutron irradiation[1–8] facilities. Estimation of the damage is also important in modern materials processing by focused ion beams, or when irradiating nanomaterials[9,10]. However, it has been recognized for several decades that the dpa value calculated from Eq. (1) for energetic cascades in pure metals on the one hand overestimates the number of stable defects by a factor of 3 to 4 (refs. [11–14]), and on the other hand underestimates the amount of atomic mixing (atoms permanently displaced from their initial lattice position to replace an atom in another position)[13,15,16] that takes place as a result of the cascade. Even though the initial effect is on the nanometric scale, it has also been estimated that it can lead to macroscopic consequences such as a 5-year underestimation of the lifetime of a nuclear reactor pressure vessel exposed to a very high thermal flux[17]. Similar trends have also been reported for intermetallic alloys[18] and ceramics[19–21]. Figure 1 illustrates the time-dependent evolution of a displacement cascade based on molecular dynamics (MD) simulations in a typical metal (see Methods section). Such a displacement cascade can be induced by a passing neutron or other high-energy (MeV or more) particle. The first lattice atom to receive a recoil energy is called the primary knock-on atom. Note how initially, when the atoms are highly excited, many of them are displaced from their lattice sites. However, as the cascade begins to thermally equilibrate with its surroundings, nearly all atoms regain positions in the perfect lattice structure. It is because of these two so-called heat spike effects[22,23] (also known in parts of the literature as 'thermal spike') that the amount of final defects generated is much smaller, and the number of atoms replacing other atoms (atomic mixing) much larger than the prediction from simple linear collision cascade models such as the NRT-dpa model (see Fig. 2). The physical reasons to this are discussed in detail in the following two subsections (building upon an earlier review work by us[24]), which also present improved functional forms and tests of these against experimental and new simulation data.

## Results

**In-cascade recombination effects on defect production**. The physical basis for the overprediction by the NRT-dpa model of the defect production at high energies is the enhanced recombination of defects in close proximity in energetic displacement cascades. The binary collision simulations used as the basis of the NRT-dpa model[3] focused on the collisional phase of the

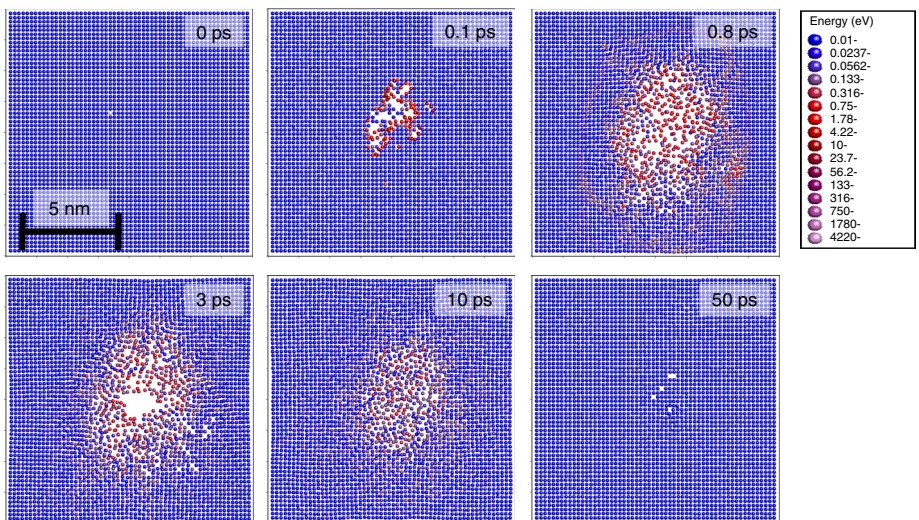

**Fig. 1** Collision cascade. A cross-sectional view of a collision cascade induced by a 10 keV primary knock-on atom in Au obtained from typical molecular dynamics simulations. The individual dots show atom positions. Blue circles illustrate atoms with low temperature and red and whitish atoms have high kinetic energies, with the energy scale given to the right. Note how initially, when the atoms are hot (high kinetic energy), a large number of atoms are displaced from their lattice sites. However, as the cascade cools down, almost all atoms regain positions in the perfect lattice sites. It is because of these two so-called 'heat spike' effects that the number of atoms replacing other atoms is much larger and the amount of final defects generated much smaller than the prediction from simple linear collision cascade models like the NRT-dpa model

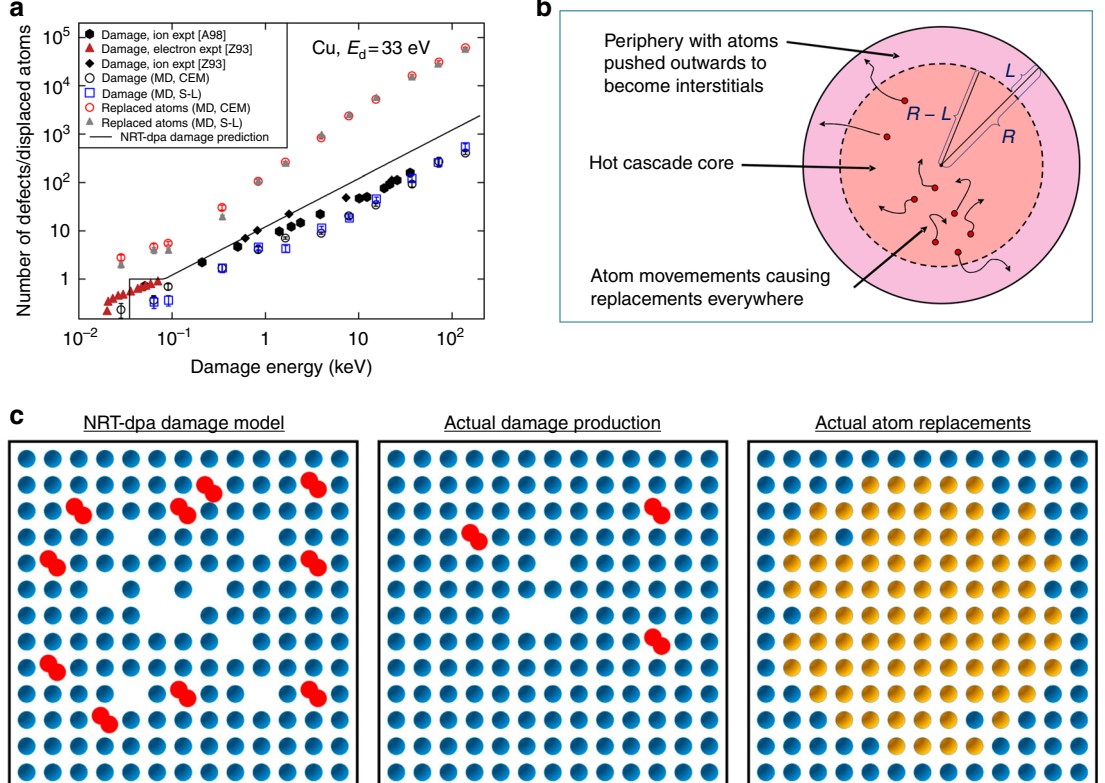

**Fig. 2** Problem with NRT-dpa. **a** Experimental and simulation data showing quantitatively the problem with the NRT-dpa equation. In the figure, 'expt' stands for experimental data, and 'MD' for simulated molecular dynamics data. The other abbreviations denoted different interatomic potentials. The references are: [A98]: ref. [26], [Z93]: ref. [13]. The Cu MD data is original work for this publication, see Methods section. The figure shows that the NRT-dpa equation does not represent correctly either the actual damage (Frenkel pairs produced) nor the number of replaced atoms. The former is overestimated by roughly a factor of 3, and the latter underestimated by a factor of 30. **b** Schematic of the concepts and quantities used in deriving the new arc-dpa and rpa equations. **c** Schematic illustration of the damage predicted by the three different damage models for the case of ~1 keV damage energy in a typical metal. For illustration purposes, the damage is illustrated as if all damage were produced in the same two-dimensional plane. Blue circles illustrate atoms in original lattice positions, yellow-brown denotes atoms that are in a different lattice position after the damage event, red atom pairs denote two interstitial atoms sharing the same lattice site, and empty lattice positions denote vacancies. Left: Damage production predicted by the NRT-dpa model. Middle: actual damage production, addressed by the new arc-dpa equation. Right: actual atom replacements, addressed by the new rpa equation, agreeing better with experimental data on number of replaced atoms (ion beam mixing). Note that in real three-imensional systems, the difference is even larger than in this 2D schematic

displacement cascade and did not consider the dynamics of cascade evolution as atomic velocities fell to the speed of sound (~5 eV) and lower, when many-body interactions become relevant. In energetically dense cascades, like that shown in Fig. 1, local melting clearly plays an important role in defect retention and structure. With increasing primary knock-on atom energy, the displacement event produces progressively more Frenkel defects (pairs of vacancies and interstitials[25]) that are spatially close to other defects. The ~10–100 jumps occurring per atom during the 1–10 ps cascade cooling phase[14] can induce significant additional recombination events as the cascade atom energies decrease, following the collisional phase, from $E_d$ to the threshold value for atomic migration ($E_m \sim 0.01$–0.3 eV for self-interstitial atoms and ~0.5–1 eV for vacancies). Accurate simulations of these cooperative multi-body effects in displacement cascades are realistically performed with MD simulations[14].

Figure 2a summarizes the defect production as a function of primary knock-on atom energy as determined from experiments performed in Cu near 4 K (where long-range thermally activated defect motion is impossible[25]). The predicted defect production and number of replaced atoms obtained from MD simulations are also shown. The figure shows that the actual defect production is

sublinear with respect to damage energy between ~0.1 and 10 keV (ref.[23]), becoming about 1/3 of the NRT-dpa prediction. At energies >10 keV corresponding to the onset of subcascade formation[14,26,27], the defect production increases linearly with damage energy but maintains the factor of ~3 lower defect production compared to the NRT-dpa value.

The physical basis for the reduction in surviving defects, with respect to the NRT model, with increasing knock-on atom energy can be understood by considering the following simplified derivation.

The ultimate survival of initially created Frenkel defects requires physical separation of the interstitial and vacancy beyond a minimum distance known as the spontaneous recombination distance (L). Atomic collisions along close-packed directions (known as recoil collision sequences) are one example of a method to efficiently transport interstitial atoms to the periphery of a displacement cascade, leaving the associated vacancy near the cascade interior. Molecular dynamics simulations[22] indicate atom transport from the displacement cascade interior may be associated with a supersonic shock-front expanding from the primary recoil event during the early stages of the cascade evolution. At low energies (below the subcascade

formation regime[14]) the displacement cascades are roughly spherical with radius $R$, and form a liquid-like zone of dense collisions (the heat spike described above).

It is further assumed that only interstitials transported to the cascade outer periphery defined by $R - L$ to $R$ will result in stable defects, whereas Frenkel pairs created in the cascade interior (0 to $R - L$) will experience recombination. The fraction of initially created NRT-dpa defects that survive is therefore given by the ratio of the outer spherical shell volume to the total cascade volume:

$$\xi_{\text{survive}} = \frac{V_{\text{outer}} - V_{\text{inner}}}{V_{\text{outer}}} = \frac{(4\pi R^3/3) - ((4\pi (R-L)^3)/3)}{4\pi R^3/3} \\ = 3\frac{L}{R} - 3\frac{L^2}{R^2} + \frac{L^3}{R^3} \approx 3\frac{L}{R} \quad (2)$$

for $L \ll R$. This 'surviving defect production fraction' $\xi_{\text{survive}}$ thus tells which fraction of defects predicted by the NRT-dpa model without any recombination survives. The cascade radius $R$ can be, within the regime of spherical cascades, estimated from classical theory of nuclear stopping power[28,29]. In practice, we used the SRIM code that implements an integral calculation to obtain mean range tables, based on cross sections from the widely used Ziegler−Biersack−Littmark (ZBL) interatomic potential[29].

We found that low-energy (less than or of the order to 10 keV) recoils of damage energy $T_d$ have an average movement distance (range) $R$ that is proportional to $T_d^x$, where the exponent $x$ is ∼ 0.4–0.6 for the metals considered in this study. Since $R \propto T_d^x$, this further gives

$$N_d'(T_d) \frac{0.8 T_d}{2E_d} \xi_{\text{survive}} = \frac{0.8 T_d}{2E_d} 3\frac{L}{R} \propto \frac{0.8 T_d}{2E_d} 3\frac{L}{T_d^x} \propto T_d^{1-x}. \quad (3)$$

This simple model thus provides an intuitive explanation for why cascade damage production is sublinear with damage energy in the heat spike regime. Physically realistic MD simulation studies[14,30] have reported that defect production rates up to the onset of subcascade formation in a variety of metals can be well described by $N_d \sim (T_d)^{1-x}$, where $x$ is between 0.2 and 0.3. These $x$ values are slightly larger than the value obtained in our simplified model because real cascades are not perfectly spherical and some defects form small clusters, reducing the recombination probability.

However, it is well known that at high energies cascades break up into subcascades[24,31,32], after which damage production

becomes linear with energy. Hence the surviving defect fraction factor $\xi(T_d)$ that accounts also for subcascade breakdown should have the feature of being a power law at low energies, but becoming a constant $c$ at high ones. A function that fulfils both criteria is

$$\xi(T_d) = A' T_d^b + c, \quad (4)$$

where $b < 0$ is consistent with the damage production efficiency reducing with increasing energy $T_d$ and the desired limit $\xi(T_d) \to c$ when $T_d \to \infty$. This thus gives a total damage production

$$N_d'(T_d) = \frac{0.8 T_d}{2E_d}\left(A' T_d^b + c\right) = \frac{0.8 A' T_d^{1+b}}{2E_d} + \frac{0.8 c T_d}{2E_d}. \quad (5)$$

Note that here the exponent $b$ is not the same as $x$, since the latter $\xi$ function is not a pure power law. The prefactor $A'$ is defined by demanding the function to be continuous, i.e. $\xi(2E_d/0.8) = 1$.

Taken together, this derivation leads us to propose (based in part on review work done within an OECD Nuclear Energy Agency group[24]) a modified defect production model, the athermal recombination corrected displacements per atom (arc-dpa).

$$N_{d,\text{arcdpa}}(T_d) = \begin{bmatrix} 0 & , & T_d < E_d \\ 1 & , & E_d < T_d < \frac{2E_d}{0.8} \\ \frac{0.8 T_d}{2E_d}\xi_{\text{arcdpa}}(T_d) & , & \frac{2E_d}{0.8} < T_d < \infty \end{bmatrix} \quad (6)$$

with the new efficiency function $\xi_{\text{arcdpa}}(T_d)$ given by

$$\xi_{\text{arcdpa}}(T_d) = \frac{1 - c_{\text{arcdpa}}}{(2E_d/0.8)^{b_{\text{arcdpa}}}} T_d^{b_{\text{arcdpa}}} + c_{\text{arcdpa}}. \quad (7)$$

Here $E_d$ is the average threshold displacement energy[33] which is the same as in the NRT-dpa and $b_{\text{arcdpa}}$ and $c_{\text{arcdpa}}$ are material constants, that need to be determined for a given material from MD simulations or experiments. The overall form (Eq. (5)) and the constant 0.8 are retained for direct comparison with the NRT-dpa model; in particular making it easy to modify computer codes that now use the NRT-dpa by simply multiplying with the function $\xi_{\text{arcdpa}}(T_d)$.

Figure 3 compares the derived arc-dpa expression for Fe and W with several recent MD simulation results used for the fitting.

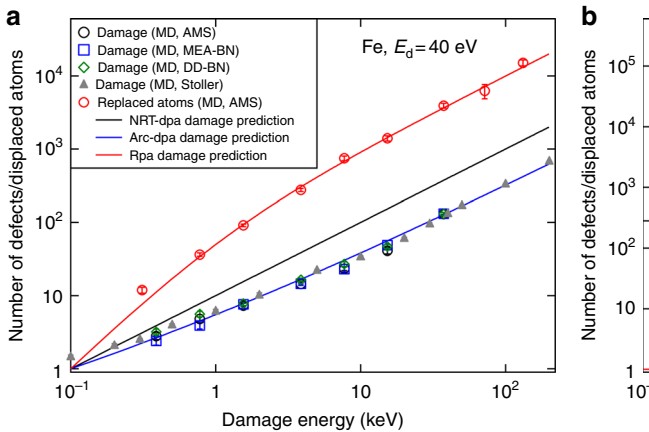
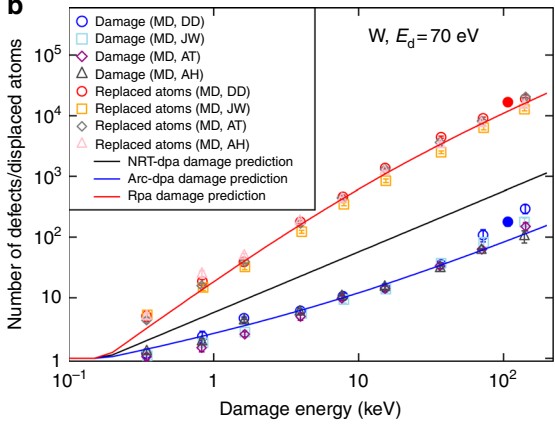

**Fig. 3** Improvement with arc-dpa and rpa. Illustration of the improvement obtained with the new arc-dpa and rpa equations for **a** Fe and **b** W. The W data also includes two data points simulated at 800 K with the DD potential (solid circles). The references are: [A98]: ref. [26], [Z93]: ref. [13]. The Fe damage data is from ref. [14] (Stoller) and ref. [48] (AMS, MEA-BN, DD-BN). The Fe replacement data and all W data is original work for this publication, see Methods section

We tested that if the fit is limited to energies <10 keV, one also can fit the data well with a power law with an exponent of ~0.7–0.8, i.e. the data is consistent with MD reports of power law dependencies. However, the arc-dpa form has the major improvement that it can also describe the saturation. Even though there is some variation in the MD data (due to differences in interatomic potentials), all of the MD results give damage production well below the $\xi = 1$ value predicted by the NRT-dpa model for cascade energies >1 keV. The arc-dpa fit to the composite data gives a reasonable averaging description of the decreasing trend in $\xi$ up to ~10 keV and the expected approach to a constant value at higher cascade energies.

**Replacements-per-atom (rpa) model**. Since the NRT-dpa model deals with production of defects that are not on perfect lattice sites, it cannot predict the number of atoms that are transported from their initial lattice site to a new lattice site, i.e. replace another atom in a perfect crystal site (right panel in Fig. 2c). This number of atom replacements is experimentally measurable via so-called radiation mixing experiments. Typically, an ion beam is used to bombard a thin marker layer inside a material, and the resulting broadening of the marker layer is measured[16,34]. For ordered alloys, it can also be conveniently measured by electrical resistivity[15]. Via an analogy with random walk atom diffusion, it is possible to relate this measured broadening to the actual number of atom replacements per ion inside the material[35]. Analysis of neutron and ion beam radiation mixing data has shown that the actual number of replaced atoms can be more than an order of magnitude larger than the number of displacements predicted by the NRT-dpa model[15,36–38]. A correct estimation of this number can be of enormous importance in predicting the effects of irradiation on phase stability[39,40] and the associated mechanical properties of materials. Nanostructured materials, such as nanolaminates and nanoscale oxide-dispersion strengthened steels, are particularly sensitive to these errors owing to their small length scales.

The superlinear increase in the number of replaced atoms with increasing knock-on atom energy can be understood by a model considering the spatial extent of a collision cascade. We consider first low energies (in the keV regime) and dense materials, where cascades are normally compact. As noted above, low-energy cascades are roughly spherical. After the ballistic phase of a cascade, MD simulations show (cf. Fig. 1) that the lattice breaks down and a liquid-like region forms. In this region all atoms are free to move and hence are almost certain to lead to one or more replacements during the thermal spike phase (as illustrated in Fig. 2c, right frame). The number of atoms $N$ in a spherical cascade of radius $R$ is proportional to the sphere volume, i.e. $N \propto R^3$, and (as already noted for the arc-dpa model) $R \propto T_d^x$. We thus find that the number of replaced atoms $N_{rpa} \propto T_d^{3x}$. Since $x > 1/3$, this simple consideration gives an intuitive explanation for why the number of replaced atoms increases superlinearly with energy at low energies, when cascades are compact. At high energies, when cascades split into subcascades[32,41], the behaviour can be expected to change to a linear dependence with damage energy. Similarly to the arc-dpa function, we thus arrive at a form

$$\xi_{rpa}(T_d) \propto \frac{T_d^{c_{rpa}}}{b_{rpa}^{c_{rpa}} + T_d^{c_{rpa}}}. \tag{8}$$

The proportionality prefactor is again set to ensure continuity, $(2E_d/0.8) = 1$. To augment the NRT equation in order to predict the number of replaced (mixed) atoms, we propose another improved damage function, the replacements-per-atom (rpa) equation. The correction factor is applied in Eq. (5) as for the arc-

dpa, but now it has the form

$$\xi_{rpa}(T_d) = \left( \frac{b_{rpa}^{c_{rpa}}}{(2E_d/0.8)^{c_{rpa}}} + 1 \right) \frac{T_d^{c_{rpa}}}{b_{rpa}^{c_{rpa}} + T_d^{c_{rpa}}}. \tag{9}$$

Here $b_{rpa}$ and $c_{rpa}$ are the new material constants. This form of $\xi_{rpa}(T_d)$ is constructed to be consistent with the derivation above. Since the NRT equation already is proportional to $T_d$, with this form the prediction is that the number of replaced atoms increases at low energies with $T_d$ as $N_{rpa} = T_d T_d^{c_{rpa}} = T_d^{1+c_{rpa}}$, i.e. $c_{rpa} = 3x - 1$. At high energies, when $T_d \gg b_{rpa}$, the form becomes linear with energy, as expected when cascades are split into subcascades. In this functional form, $b_{rpa}$ has a physical meaning as the average energy for subcascade breakdown in terms of number of replaced atoms. Moreover, similar to the arc-dpa form, Eq. (9) fulfils the same conditions of continuity and compatibility with the NRT-dpa model.

**Discussion**

We first reiterate the physical meaning of the newly introduced material constants. $b_{rpa}$ is related to the subcascade formation energy, and has energy units. The unitless exponents $b_{arcdpa}$ and $c_{rpa}$ are associated with the dependence of the ion range with energy. Finally, the unitless quantity $c_{arcdpa}$ is associated with the saturation value of damage recombination with heat spike size.

As a summary of the arc-dpa and rpa models, Fig. 3 compares the obtained rpa and arc-dpa curves with the NRT dpa prediction for Fe and W, as a function of damage energy. Note that the NRT-dpa damage production equation does not, except at the very lowest energies near 100 eV, describe correctly either the surviving defects or the amount of radiation mixing in energetic displacement cascades. The newly introduced arc-dpa correction factor $\xi_{arcdpa}$ for the primary damage leads to a calculated surviving defect fraction about a factor of 3 lower than the NRT-dpa prediction, and the rpa correction factor $\xi_{rpa}$ leads to the calculated number of atom replacements to be about 30 times higher than the predicted NRT displacement value. Both the correction factors agree very well with MD simulation results over more than three orders of magnitude in energy for all elements, giving confidence that the derived functional forms are well motivated. We note that when additional and more accurate MD or experimental data becomes available, the models (Eqs. (6) and (9)) could be refined for a better description e.g. near the threshold.

We also considered the dependence of the results on the ambient temperature. Several previous studies have shown that the effect of ambient temperature on primary damage production or atom replacements at ps time scales is insignificant or weak up to temperatures around roughly half the melting point[14,42,43]. For this work, we also simulated two of the data points for W at an elevated temperature, 800 K. The results (solid circles in Fig. 3b) show that both the damage and replacements is the same within the statistical uncertainty as those at low temperature for the same potential. We note that given sufficiently large and statistically accurate data sets for a range of higher elevated temperatures, it would be possible to make the arc-dpa and rpa model parameters temperature-dependent.

Table 1 gives results of the arc-dpa and rpa model fit parameters for several metals based on MD data. The metals were chosen as those for which a sufficiently wide MD database was available for the fitting.

We note that, in spite of its failure to predict damage production correctly, the original NRT-dpa standard remains useful for comparing scaled radiation dose (exposure), as it is essentially proportional to the collision-relevant portion of radiation energy deposited per volume. The correct use of this standard as the first

**Table 1 Material constants**

| Material | $E_d$ (eV) | $b_{arc-dpa}$ | $c_{arc-dpa}$ | $b_{rpa}$ (eV) | $c_{rpa}$ |
|---|---|---|---|---|---|
| Fe | 40 | −0.568 ± 0.020 | 0.286 ± 0.005 | 1018 ± 145 | 0.95 ± 0.04 |
| Cu | 33 | −0.68 ± 0.05 | 0.16 ± 0.01 | 3319 ± 249 | 0.97 ± 0.02 |
| Ni | 39 | −1.01 ± 0.11 | 0.23 ± 0.01 | 3325 ± 230 | 0.92 ± 0.01 |
| Pd | 41 | −0.88 ± 0.12 | 0.15 ± 0.02 | 2065 ± 183 | 1.08 ± 0.02 |
| Pt | 42 | −1.12 ± 0.09 | 0.11 ± 0.01 | 5531 ± 762 | 0.87 ± 0.02 |
| W | 70 | −0.56 ± 0.02 | 0.12 ± 0.01 | 12,332 ± 1250 | 0.73 ± 0.01 |

Results for the arc-dpa and rpa material constants for a number of metals. The errors are given in s.e.m.

variable by which to compare radiation damage levels in different environments remains, therefore, strongly recommended. The new models, arc-dpa and rpa, developed here extend the usefulness of the dpa in that, while they can also be used to compare different irradiations, they in addition give accurate predictions of primary damage production and radiation mixing. They are deliberately constructed in such a way that they are easy to implement in existing radiation effects software: they add only four additional parameters (or two, if only one of the two new models is used) for each element. The arc-dpa and rpa equations thus enable a significant increase in physical relevance with minimal increase in computational efficiency or complexity. Of course, however, the practical application of arc-dpa and rpa requires the construction of suitable databases of parameters for each material: not only $E_d$ but also the $b$ and $c$ constants. Collectively, the new models represent an important step towards improved quantification of the primary damage state during irradiation of materials.

In general, these damage models are expected to be relevant for many other materials besides the elemental metals discussed here. Studies of damage in metal alloys indicate the arc-dpa model will be directly applicable to both dilute[44] and concentrated[45] metal alloys. In non-metallic materials, the arc-dpa function may not be universally relevant, as damage production involves effects such as amorphization[25] that cannot be captured by any simple equation. However, some ceramic materials are known to undergo significant in-cascade recombination[19–21], and for these arc-dpa can be useful. On the other hand, the rpa function can be expected to be relevant in any material where heat spikes are significant (i.e. all dense materials), since in all of these the atomic mixing will be enhanced by collective atomic motion. The formation of the arc-dpa and rpa also motivate systematic experimental and simulation studies to understand better the primary state of damage in non-metallic systems, where (with the exception of Si) studies are scarce.

Prior to concluding, we emphasize that the arc-dpa and rpa models deal with the primary damage state only, i.e. the damage produced during the first few ps after a collision cascade initiated. Already at room temperature, thermally activated defect migration is known to be significant, and can reduce the damage production significantly from the arc-dpa value due to recombination effects, or enhance atom mixing from the rpa value. They also do not describe defect clustering or damage overlap effects[45–47]. However, even for these cases the new functions can be useful, as a starting point for e.g. kinetic Monte Carlo or rate theory calculations of high-dose irradiation effects[45] (where cascades overlap) or conditions where thermal defect migration recombines defects.

In conclusion, the new arc-dpa and rpa models introduced here allow, in a very simple and efficient way, to incorporate the improved understanding on radiation defect generation mechanisms gained during the last four decades into software calculating primary radiation damage generation rates in macroscopic reactor components. The arc-dpa model accounts for the enhanced recombination active in pure metals as well as in many alloys, which strongly reduces the number of point defects present in the primary damage, compared to the traditional NRT-dpa model. The rpa model provides a measure of the volume of the irradiated material directly affected by the cascade, which is important e.g. for phase stability considerations. In calculations of radiation damage effects where the dpa measure is used as a starting point, these new functions provide improved accuracy in a simple analytical form. This allows, e.g., differentiating between irradiation conditions dominated by either low- or high-energy recoils, and perhaps even more importantly, introduces the possibility to quantify analytically the very large (about two orders of magnitude) difference between damage production and atom relocation effects.

## Methods

**Molecular dynamics simulations.** The new data used for the fits of the arc-dpa and rpa functions were obtained from MD simulations following Refs. [31,36,37,48,49]. The Fe damage data is from ref. [48] or the references indicated in the figure, and the Ni, Pd and Pt data from ref. [49], and other data previously unpublished. In all cases, a crystalline simulation cell in either the face-centred cubic or body-centred cubic crystal structure was first constructed and equilibrated by a short (few picoseconds) MD simulation at room temperature. The interatomic interactions were modelled with equilibrium reactive interatomic potentials to which the ZBL repulsive potential[36] was joint at small separations to realistically mimic high-energy interactions. The interatomic potentials indicated by abbreviations in the figures are for Cu: CEM: ref. [50], S-L: ref. [51] and for W: DD: ref. [52], JW: ref. [53], AT: ref. [54], AH: ref. [55].

Periodic boundary conditions were used in all directions, to correspond to high-energy ion or neutron effects deep inside a material, and the lattice constant was set to the equilibrium value at 0 K (previous works show that 0 and 300 K primary damage results are identical within the statistical uncertainty in transition metals). After equilibration, an atom was selected randomly near the centre of the simulation cell, and given a recoil energy in a random direction in three dimensions. The central parts of the simulation cell were simulated in the NVE ensemble, while excess energy was removed from the system using temperature scaling towards room temperature in the outermost 1 unit cell thick regions of the cell. To account for energy loss to ionizations, the ZBL96 electronic stopping power[29] was applied as a frictional force on all atoms with a kinetic energy higher than 10 eV, and the damage energy $T_d$ was calculated as the difference between the initial recoil energy and the total sum of energy lost to electronic stopping. The damage was analysed using the Wigner−Seitz cells approach[31] that is space-filling and hence allows for a unique determination of whether a defect is vacancy- or interstitial-type.

**Data availability.** The new molecular dynamics data that has been produced for this paper, and comprises the source data for the parameters given in Table 1, is freely available for download at http://urn.fi/urn:nbn:fi:csc-kata20180125132021651079.

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

## Acknowledgements

We thank the OECD/NEA for setting up the initial working group that initiated this work. This work was sponsored in part by the U.S. Department of Energy, Office of Fusion Energy Sciences, SJZ, RES, Office of Basic Energy Sciences, RSA, WJW. A.E.S. acknowledges support from the Academy of Finland through project No. 311472. Grants of computer time from CSC—the Finnish IT Center for Science as well as the Finnish Grid and Cloud Infrastructure (persistent identifier urn:nbn:fi:research-infras-2016072533) are gratefully acknowledged. This work has been also partly carried out within the framework of the EUROfusion Consortium and has also received partial funding from the Euratom research and training programme 2014–2018 under grant agreement no. 633053. The views and opinions expressed herein do not necessarily reflect those of the European Commission.

## Author contributions

K.N. was the lead author of the theory and modelling parts, guided the computational simulations, derived the rpa equation, and did the fitting of the arc-dpa and rpa models to the plots. S.J.Z. derived the arc-dpa equation, performed the analyses of experimental data and stimulated activities on the importance of rpa. A.E.S. simulated all the new MD data. R.S.A. and S.J.Z. were lead authors on the discussion of experiments and R.S.A on the mixing parts. All authors contributed to data interpretation and to the manuscript preparation and approved the final version of the manuscript.

## Additional information

**Competing interests:** The authors declare no competing interests.

