## [Peer Review File · Nature Communications]

Reviewers' Comments:

Reviewer #1:

Remarks to the Author:

In this manuscript, the authors propose a new way of quantifying the damage produced in metals by energetic particles that goes beyond the well-known and widely used model of Norgett, Robinson and Torrens, the NRT-displacements per atom model. The NRT-dpa was introduced to be able to quantitatively compare irradiation effects between different materials and irradiation sources. Although the failures of this model have been known for decades, no other model to address these issues existed until now. The authors have been able to achieve this significant contribution thanks to the knowledge acquired over the years and the existing database of collision cascades in metals. In fact, this is the result of a large collaboration between some of the major research centres working on radiation damage effects. Although the new models proposed here have been derived for metals, they show a path for extending it to other materials such as semiconductors or ceramics. Therefore, this work is an important breakthrough in the field of radiation effects and defect production by energetic particles in general that will influence how damage is quantified in irradiated materials. It shows a significant advancement in a topic of relevance to researchers working in several fields, which justifies its publication in Nature Communications.

The authors however need to make some changes and clarify a few points in the manuscript before it can be accepted for publication. The following are some general remarks and some minor changes that must be addressed.

Introduction: when talking about the threshold displacement energy, there is a sentence in page 1 "Ed, estimated to be on the order of 25 eV" and then this value is mentioned again in page 2 "Typical values of Ed for different materials range from 20 to 100 eV". These two descriptions or values should be grouped or the difference explained. As it is written now, it could lead to confusion.

Page 2. Second paragraph, instead of 'crystalline materials' it might be more accurate to say 'solids' since the dpa can also be used for amorphous materials.

Some terms are used that might not be known for a wider audience, not directly involved in radiation damage but interested in these findings. For example, the term primary knock-on atom is first used in page 3. This should be introduced when describing figure 1 in page 2. The energy of the primary knock-on atom used for the results of figure 1 should also be given. Frenkel defects is another term also used, that could be described when mentioning the number of displacements in page 2.

On arc-dpa and the comparison to experimental data. The MD simulations provide a very good agreement with the experimental measurements in copper, and much better than the NRT-dpa model except for low energies (below 1keV) and in particular when comparing to the electron irradiation data. Do the authors have any explanation for this discrepancy?.

Page 3. Last paragraph: " ... in Cu near 4 K (where long range thermally activated defect motion is impossible)", probably more accurate 'unlikely' or 'very rare'.

Page 6: Sentence "Analysis of neutron and ion beam radiation mixing data has shown that the actual number of replaced atoms can be more than an order of magnitude larger than the number predicted by the NRT-dpa model", to avoid confusion, it should say the number of displacements predicted by the NRT-dpa model.

In the discussion, the authors should strengthen more that this is the damage produced within the first few picoseconds of an irradiation and that many more processes occur that are not captured

within this model since they involve complex phenomena such as defect diffusion, clustering, interaction with the pre-existing microstructure and other effects.

Also regarding the discussion, the NRT-dpa model should fail at low energies ($\sim 5\text{eV}$) where many-body interactions are relevant. But it actually provides a good description of the damage at the lowest energies. This seems to be a contradiction. Could the authors comment on this?. Could the density of low energy events have something to do with this?

Figures:

Figure 1 needs better resolution since it is difficult to read the axis. Instead, the dimensions of the box could be given. The description of the figure caption regarding the colours in the figure does not correspond to the labels in the figure. Are they coloured by energy or by location? Probably those atoms with higher energy are displaced from their original positions and the two descriptions are equivalent but not necessarily. In fact, at the end all atoms show up as blue but probably many of them are not in their original positions (but back into a lattice site). The correct description of the colour coding used should be given both in the text and in the figure caption. Also, maybe a more proper description should be given in this figure caption: "... a lot of atoms are displaced from their lattice sites", a high percentage of the atoms in the simulation box, a large number of atoms in the simulation box ...

Figure 2: there is a mistake in figure 2a. Only blue and red atoms are shown, but no yellow_brown, as mentioned in the text. So the figure for the rpa model shows no changes from a perfect lattice.

The data for Ni in figure 3, where does it come from? If it is from Ref. 42, as mentioned in the methods section, the reference should be included here as well.

Finally, as a suggestion, the authors might want to revisit the first part of the introduction to highlight the interest of this work in current applications, adding some references to recent reviews in nuclear energy applications or semiconductors, as well as nanomaterials. Focused ion beams, for example, are now used for nano-patterning surfaces and other applications, and these results could be helpful in this field as well. This would make the manuscript appealing to a broader audience.

Reviewer #2:

Remarks to the Author:

The paper describes a modification of the Norgett-Robinson-Torrens displacements per atom (NRT-dpa) model, which is known to underestimate the number of defects and neglect the replacement events (atomic mixing) that could be significantly more efficient than the defect formation rate. The paper is relatively well written, the simple ideas that are put forward are clearly explained, however the research lacks the significant advancement factor and high quality aspect. In particular, the proposed model, or correction to the NRT-model, is based and depends on the molecular dynamics simulations data. The four fitting parameters have to be fitted to these data. As shown in Table 1. these coefficients vary a lot between different metals and are far from being unique. To me, it sounds more like a fitting function than a model, although somehow justified, or than a general solution to the problem with the NRT-dpa model. A significant shortcoming of this studies is that Authors neglect the effect of temperature, which should promote the recombination as shown in other molecular dynamics studies (e.g. Robinson et al. Phys. Rev. B 86, 134105 (2012)). Below I give more details and comments.

Detailed comments:

1) The presented results and model are sound and reasonably well justified, although the scientific

content lacks the significant advance factor and high quality of the research. It for sure would be adequate for a regular journal in the respected field, but does not fit into the aims and scope of Nature Communications.

2) The proposed model, or correction to the NRT-model, is based and depends on the molecular dynamics simulations data. The four fitting parameters have to be fitted to these data. As shown in Table 1. these coefficients vary a lot between different metals and are far from being unique. To me, it sounds more like a fitting function than a model, although somehow justified.

3) Authors do not mention what is the effect on temperature. It is known from various MD studies that temperature enhances recombination and defect formation. See for instance Robinson et al. Phys. Rev. B 86, 134105 (2012). The effect of temperature should be clearly discussed. It is hard to believe that temperature is not affecting the defect formation/recombination rate in MD simulations and that the coefficients presented in Table 1 are temperature-independent.

4) It is also known that damage extent as simulated by MD does not have to follow linear trend as a function of energy. See for instance the defect formation probabilities as a function of PKA energy (again, Robinson et al. Phys. Rev. B 86, 134105 (2012)). Such studies, and there is quite a few similar ones, should be discussed by the authors.

5) If Authors want to convince the readers that their model represents significant advancement in the fields, they should test it on a larger set of data (MD, experimental) and on more complex materials. Six cases is not "several" as claimed in the section 4. Otherwise, corrections (3) and (4) can be seen just as a fitting functions that allow for reconciling prediction of a simple NRT-dpa model with the MD data, but not as a general solution to the problem

6) Authors benchmark on classical MD simulations results. How would the result change if for instance ab initio MD simulation results would be available? This should be explained.

Reviewer #3:

Remarks to the Author:

The question of how to estimate radiation damage in solids is of long-standing interest. This article presents an update on the rather simplistic NRT model which is widely used, but not theoretically well grounded. It is well known in the community that NRT is simply a starting point, and its benefits are predominantly to provide order of magnitude estimates and compare relative effects.

On the surface this article presents an advance on the NRT model, by providing some physical scaling arguments and fitting to data from Molecular Dynamics (MD). However, the arguments presented are not particularly convincing, particularly for the so-called "arc-dpa" model. In its present form, this manuscript would not be suitable for a discipline-specific journal, let alone one of the status of Nature Communications.

In no particular form, the deficiencies of this article are:

1) The title contains acronyms which make no sense to the reader without first reading the manuscript

2) Figure 1 serves no purpose with regards the new models, and simply presents what is well known in the field. Anyone with an interest in revised NRT models would be well aware of this.

3) Figure 2 is confusing and poorly thought out; Figure 2c should be grouped with Figure 3c & 3d. Figure 2b has lots of data (MD+ experiment) but the legend is a mishmash of acronyms. It presents the NRT model data, but not anything from the new model. Figure 2a is a separate concept. The

correct conceptual order is i) Figure 2b (explains the problem) , (ii) Figure 2a (i.e. schematic of the new model), iii) Figure 2c + Figure 3.

4) The explanations of the physical basis of the revised form is incredibly hard to read. Even someone well-versed in the field will struggle to wade through the various scaling arguments. Even if the the description were to be substantially improved, the basic functional form for arc-dpa does not seem to be correct. As T_d increases, the functional form should reduce to the linear form as noted at the bottom of page 3. However, this limiting case is not built into the functional form. For this reason alone the paper should be rejected, as the argument for a physical basis is thrown out the window at the curve fitting stage. The parameter fitting is simply an exercise in fitting data with any old function that works. There is nothing wrong with blind fitting of course, but that is not what is being claimed here. Inspection of the arc-dpa quantities in Table 1 shows that none of the scaling arguments survive into the actual functional form.

4a) Note that section 3 does not have the same problems as noted in point (4). Here the functional form has the correct trend for large T_d and the fitted constants have a physically meaningful interpretation.

5) There is no mention that the E_d values themselves are often open to interpretation. Certain values have become commonly used, but given that NRT overestimates damage by roughly constant factors, the obvious work-around is to use a higher E_d , or alternatively, use a factor other than 0.8 in Equation 1. It is well-known that the 0.8 is basically a "fudge-factor" which recognizes that the Kinchin-Pease model predicts too many defects.

To summarise, the authors are right to attempt to improve on the NRT model, but the present manuscript is not a major enough advance to warrant publication. The heart of the manuscript is essentially an exercise in curve fitting, which could be equally achieved using cubic splines or similar functions. There is no attempt to extract further information based on the parameters, and in the case of Equation 3 the argument developed in the text does not translate to the proposed functional form.

We thank the referees for their careful reading of the manuscript. In addition to the general motivation given on the previous page, we provide below a detailed response to the referee's questions, including a description of a more detailed derivation of the arc-dpa model that we believe clarifies the technical and quality concerns of the referees.

The original report is given in *blue courier italic* font, and our response in regular black text. To be able to cross-cite references, we added a numbering to the referee comments and answers: R1.1, A.1.1 etc. The main new texts added to the manuscript are provided in **red font**.

Sincerely yours,

The authors

Reviewers' comments:

Reviewer #1 (Remarks to the Author):

R.1.1. In this manuscript, the authors propose a new way of quantifying the damage produced in metals by energetic particles that goes beyond the well-known and widely used model of Norgett, Robinson and Torrens, the NRT-displacements per atom model. The NRT-dpa was introduced to be able to quantitatively compare irradiation effects between different materials and irradiation sources. Although the failures of this model have been known for decades, no other model to address these issues existed until now. The authors have been able to achieve this significant contribution thanks to the knowledge acquired over the years and the existing database of collision cascades in metals. In fact, this is the result of a large collaboration between some of the major research centres working on radiation damage effects. Although the new models proposed here have been derived for metals, they show a path for extending it to other materials such as semiconductors or ceramics. Therefore, this work is an important breakthrough in the field of radiation effects and defect production by energetic particles in general that will influence how damage is quantified in irradiated materials. It shows a significant advancement in a topic of relevance to researchers working in several fields, which justifies its publication in Nature Communications.

A.1.1. We thank the referee for this very positive overall judgment.

R1.2. The authors however need to make some changes and clarify a few points in the manuscript before it can be accepted for publication. The following are some general remarks and some minor changes that must be addressed.

Introduction: when talking about the threshold displacement energy, there is a sentence in page 1 E_d , estimated to be on the order of 25 eV; and then this value is mentioned again in page 2 E_d . Typical values of E_d for different materials range from 20 to 100 eV;. These two descriptions or values should be grouped or the difference explained. As it is written now, it could lead to confusion.

A1.2. We thank the referee for pointing out this apparent discrepancy. We now simply removed the initial mention of the 25 eV, since this was not in any case necessary as the wider range is mentioned just a few sentences below.

R1.3. Page 2. Second paragraph, instead of ‘crystalline materials’; it might be more accurate to say ‘solids’; since the dpa can also be used for amorphous materials.

A1.3. The referee is perfectly correct, and the text has now been modified as suggested.

R1.4. Some terms are used that might not be known for a wider audience, not directly involved in radiation damage but interested in these findings. For example, the term primary knock-on atom is first used in page 3. This should be introduced when describing figure 1 in page 2. The energy of the primary knock-on atom used for the results of figure 1 should also be given. Frenkel defects is another term also used, that could be described when mentioning the number of displacements in page 2.

A1.4. We thank the referee for these constructive comments. We have now changed the manuscript according to the suggestions, with changes marked in red.

R1.5. On arc-dpa and the comparison to experimental data. The MD simulations provide a very good agreement with the experimental measurements in copper, and much better than the NRT-dpa model except for low energies (below 1keV) and in particular when comparing to the electron irradiation data. Do the authors have any explanation for this discrepancy?.

A.1.5. We do not have a definite explanation to this. It is clear (also from the rpa model comparison with MD data) that the arc-dpa and rpa form are not fully suitable for describing the lowest energies, near the threshold. To retain backward compatibility in the models, we chose, however, to retain the step-like function at the threshold. An additional reason to this is that neither all experimental data nor MD simulations are consistent in the low-energy regime, and hence it would be premature to extend the functional form. We emphasize, however, that the regime most relevant to neutron damage in nuclear reactors is the high-energy one, where both experimental and MD data are fairly consistent with each other.

R1.6. Page 3. Last paragraph: ‘;’; in Cu near 4 K (where long range thermally activated defect motion is impossible);, probably more accurate ‘unlikely’; or ‘very rare’;.

A1.6. We are actually certain that there is no long-range migration, as this temperature regime is below so called “Stage I”, which means that no defect motion occurs. We added a reference to this experimental observation.

R.1.7 Page 6: Sentence ‘Analysis of neutron and ion beam radiation mixing data has shown that the actual number of replaced atoms can be more than an order of magnitude larger than the number predicted by the NRT-dpa model’;, to avoid confusion, it should say the number of displacements predicted by the NRT-dpa model.

A1.7. Corrected as suggested.

R1.8. In the discussion, the authors should strengthen more that this is the damage produced within the first few picoseconds of an irradiation and that many more processes occur that are not captured within this model since they involve complex phenomena such as defect diffusion, clustering, interaction with the pre-existing microstructure and other effects.

A1.8. We now added a new paragraph, just before the conclusions, discussing this:

Prior to concluding, we do emphasize that the arc-dpa and rpa models deal with the primary damage state only, i.e. the damage produced during the first few ps after a collision cascade initiated. Already at room temperature, thermally activated defect migration is known to be significant, and can reduce the damage production significantly from the arc-dpa value due to recombination effects, or enhance atom mixing from the rpa value. They also do not describe defect clustering or damage overlap effects⁴². However, even for these cases the new functions can be useful, as a starting point for e.g. kinetic Monte Carlo or rate theory calculations of high-dose irradiation effects⁴² (where cascades overlap) or conditions where thermal defect migration recombines defects.

R1.9 Also regarding the discussion, the NRT-dpa model should fail at low energies (~5eV) where many-body interactions are relevant. But it actually provides a good description of the damage at the lowest energies. This seems to be a contradiction. Could the authors comment on this?. Could the density of low energy events have something to do with this?

A1.9. We already discussed the complexities of the low-energy effects above, see answer A1.5. . The step function of all these models (NRT-dpa, arc-dpa and rpa) is known not to be fully realistic, but as we noted above, it may be premature to develop a new functional form either, as the reference data is not fully consistent.

Figures:

R1.10 Figure 1 needs better resolution since it is difficult to read the axis. Instead, the dimensions of the box could be given. The description of the figure caption regarding the colours in the figure does not correspond to the labels in the figure. Are they coloured by energy or by location? Probably those atoms with higher energy are displaced from their original positions and the two descriptions are equivalent but not necessarily. In fact, at the end all atoms show up as blue but probably many of them are not in their original positions (but back into a lattice site). The correct description of the colour coding used should be given both in the text and in the figure caption. Also, maybe a more proper description should be given in this figure caption: 'A lot of atoms are displaced from their lattice sites', a high percentage of the atoms in the simulation box, a large number of atoms in the simulation box ...

A1.10. Indeed the color scale description was misleading, as noted by the referee. We now changed the caption as suggested. We also edited the figure itself for better readability, as suggested.

R1.11 Figure 2: there is a mistake in figure 2a. Only blue and red atoms are shown, but no yellow/brown, as mentioned in the text. So the figure for the rpa model shows no changes from a perfect lattice.

A.1.11.

This must be a technical issue on the colour scale, as in the version on our screen, the rpa figure on the right does have yellow/brown atoms. We still reproduce it here for the referee's convenience, with a black circle indicating the region with yellow/brown atoms.

We would be happy to work with the technical editors to make sure the colour scale is correct in the final manuscript.

R1.12. The data for Ni in figure 3, where does it come from? If it is from Ref. 42, as mentioned in the methods section, the reference should be included here as well.

A1.12 Indeed the Ni data is from Ref. 42. On the other hand, due to changes requested by referee 3, we not removed the Ni figure.

R1.13 Finally, as a suggestion, the authors might want to revisit the first part of the introduction to highlight the interest of this work in current applications, adding some references to recent reviews in nuclear energy applications or semiconductors, as well as nanomaterials. Focused ion beams, for example, are now used for nano-patterning surfaces and other applications, and these results could be helpful in this field as well. This would make the manuscript appealing to a broader audience.

A1.14. We thank the referee for this good suggestion. We now modified the introduction by adding a sentence along these lines.

Reviewer #2 (Remarks to the Author):

R2.1. The paper describes a modification of the Norgett-Robinson-Torrens displacements per atom (NRT-dpa) model, which is known to underestimate the number of defects and neglect the replacement events (atomic mixing) that could be significantly more efficient than the defect formation rate. The paper is relatively well written, the simple ideas that are

put forward are clearly explained, however the research lacks the significant advancement factor and high quality aspect. In particular, the proposed model, or correction to the NRT-model, is based and depends on the molecular dynamics simulations data. The four fitting parameters have to be fitted to these data. As shown in Table 1. these coefficients vary a lot between different metals and are far from being unique. To me, it sounds more like a fitting function than a model, although somehow justified, or than a general solution to the problem with the NRT-dpa model. A significant shortcoming of this studies is that Authors neglect the effect of temperature, which should promote the recombination as shown in other molecular dynamics studies (e.g. Robinson et al. Phys. Rev. B 86, 134105 (2012). Below I give more details and comments.

A2.1. The paper referred to by the referee deals with thermal effects in an oxide (rutile TiO₂) at low energies (well below 1 keV), which is a quite different from the focus of this work. As we state in the paper, radiation effects in ionic materials are complicated, and in this work we do not aim to systematically go through for which oxides the arc-dpa and rpa models may be relevant.

The energy range in the cited paper is also beyond the focus of this paper, where the main aim is to obtain a better description in the heat spike regime, above 1 keV.

In the focus area of this paper, heat spike regime in metals, the temperature effect on primary damage production has been studied by several of the authors, e.g. Stoller et al, J. Nucl. Mater. 276 (2000) 22. This work showed that up to 600 K, the effect of temperature on damage production in heat spikes is quite small. The reason is that the recombination (also discussed in this paper) in the very hot heat spikes dominates the final damage production,

Temperature effects are certainly of some importance on long time scales, and we did consider this in our work. However, we concluded from review of recent work in the field that it is not possible to make a single analytical model for a given material to describe thermal recombination, as this becomes a balance of defect cluster size, temperature and material microstructure. We now added (see answer A1.8. to the first referee) an explanation that the current models can, however, be used as a starting point for thermal effect modelling by e.g. Kinetic Monte Carlo or rate theory approaches.

Detailed comments:

R2.1 1) The presented results and model are sound and reasonably well justified, although the scientific content lacks the significant advance factor and high quality of the research. It for sure would be adequate for a regular journal in the respected field, but does not fit into the aims and scope of Nature Communications.

A2.1. Here we refer to our initial letter to the editor and the statements of referee 1, which motivate the broad importance of our work.

2) The proposed model, or correction to the NRT-model, is based and depends on the molecular dynamics simulations data. The four fitting parameters have to be fitted to these data. As shown in Table 1. these coefficients vary a lot between different metals and are far from being unique. To me, it sounds more like a fitting function than a model, although somehow justified.

A2.2. We do not understand this comment by the referee. Saying that material properties should be describable with a single set of parameters is like saying all materials in nature should be identical to fit beautifully a mathematical form made by man. In reality, of course, it is extremely well established that

real materials are different: they have different colour, melting point, density, crystal structure, bulk modulus, ... To describe this scientifically, one needs a set of **material constants**. Because materials are different in their basic properties, it is also natural that their radiation properties are. Hence the new constants we introduce can be considered new material constants for radiation effects. To emphasize this in the text, we now changed the misleading term “fitting constant” with “material constant”. Moreover, they are not without physical meaning: the constant b_{rpa} is clearly a measure of subcascade breakdown energy (as stated in the paper), and $c_{arc-dpa}$ is the saturation level of defect recombination. The exponents $b_{arc-dpa}$ and C_{dpa} are related to the defect recombination and heat spike cooling times, respectively.

R2.3 3) Authors do not mention what is the effect on temperature. It is known from various MD studies that temperature enhances recombination and defect formation. See for instance Robinson et al. Phys. Rev. B 86, 134105 (2012). The effect of temperature should be clearly discussed. It is hard to believe that temperature is not affecting the defect formation/recombination rate in MD simulations and that the coefficients presented in Table 1 are temperature-independent.

A2.3. See response A2.1 above.

R2.4 4) It is also known that damage extent as simulated by MD does not have to follow linear trend as a function of energy. See for instance the defect formation probabilities as a function of PKA energy (again, Robinson et al. Phys. Rev. B 86, 134105 (2012)). Such studies, and there is quite a few similar ones, should be discussed by the authors.

A2.4 Yes, this is of course the case at the low energy regime near the threshold, and also we have studied this effect (see e.g. Tarus et al, Phys. Rev. B **58**, 9907 (1998) and Nordlund et al, Nucl. Instr. Meth. Phys. Res. B **246**, 322 (2005). However, the point of the arc-dpa and rpa models is exactly that they are not linear with energy in the low-energy regime. The paper by Robinson et al extends only up to 200 eV, and in this regime indeed our models are not linear. The requirement for a linear behavior we discuss in the paper only comes in at energies around 100 keV due to subcascade breakdown, which is an effect not at all discussed in the Robinson paper.

R.2.5. 5) If Authors want to convince the readers that their model represents significant advancement in the fields, they should test it on a larger set of data (MD, experimental) and on more complex materials. Six cases is not "several" as claimed in the section 4. Otherwise, corrections (3) and (4) can be seen just as a fitting functions that allow for reconciling prediction of a simple NRT-dpa model with the MD data, but not as a general solution to the problem

A2.5. The recombination behavior and enhanced mixing discussed in this paper have been consistently observed in all dense metals studied with respect to radiation effects, including both dilute and concentrated metal alloys, and this is already discussed in the paper. This forms a huge body of technologically important materials. Hence the arc-dpa and rpa models will certainly be valid in all of these. We emphasize that this paper is the first one on the models themselves, and hence it is beyond the point of the paper to present parameters for a wide range of metals. As discussed in the paper, we hence chose to give the parameters only for those metals for which several consistent data sets are available.

R2.6. 6) Authors benchmark on classical MD simulations results. How would the result change if for instance ab initio MD simulation results would be available? This should be explained.

A2.6. Although simulation of collision cascades in the heat spike regime is still beyond the capability of MD simulations, we do not expect that they would give significantly different results. The reason to this statement is that completely differently developed classical potentials do give consistent damage production numbers in the heat spike regime [Björkas et al, Nucl. Instr. Meth. Phys. Res. B 259, 853 (2007)]. The reason is that the crucial damage recombination occurs at the recrystallization front, and the velocity of this is largely determined by the melting point. Hence any model (classical or quantum mechanical) that gives a good description of the melting point, should give roughly similar damage numbers in the heat spike regime.

In the near-threshold regime, there are recent ab initio calculations available of radiation effects. They do show that classical potentials and the ab initio calculations give fairly similar results [Olsson et al, Mater. Res. Lett. 4 (2016) 219]. The reported difference of about 20% in there is much smaller than the factor of 3 or 30 differences to NRT-dpa addressed by the arc-dpa and rpa models.

Reviewer #3 (Remarks to the Author):

R3.1. The question of how to estimate radiation damage in solids is of long-standing interest. This article presents an update on the rather simplistic NRT model which is widely used, but not theoretically well grounded. It is well known in the community that NRT is simply a starting point, and its benefits are predominantly to provide order of magnitude estimates and compare relative effects.

On the surface this article presents an advance on the NRT model, by providing some physical scaling arguments and fitting to data from Molecular Dynamics (MD). However, the arguments presented are not particularly convincing, particularly for the so-called ‘arc-dpa’ model. In its present form, this manuscript would not be suitable for a discipline-specific journal, let alone one of the status of Nature Communications.

A3.1. We strongly disagree with this statement, which seems to come from a misunderstanding of the derivation of the arc-dpa form. We do admit, however, that this derivation was rather terse in the original manuscript, and hence could be misunderstood. We now extended the description of the model in the manuscript, and provide here an even more detailed form for the benefit of the referee:

The ultimate survival of initially created Frenkel defects requires physical separation of the interstitial and vacancy beyond a minimum distance known as the spontaneous recombination distance (L). Atomic collisions along close-packed directions (known as “recoil collision sequences”) are one example of a method to efficiently transport interstitial atoms to the periphery of a displacement cascade, leaving the associated vacancy near the cascade interior. Molecular dynamics simulations²² indicate atom transport from the displacement cascade interior may be associated with a supersonic shock-front expanding from the primary recoil event during the early stages of the cascade evolution. Recognizing that at low energies (below subcascade formation regime¹⁴) the displacement cascades are roughly spherical with radius R , forming a liquid-like zone of dense collisions (the heat spike described above).

It is further assumed that only interstitials transported to the cascade outer periphery defined by $R-L$ to R will result in stable defects, whereas Frenkel pairs created in the cascade interior (0 to $R-L$) will experience recombination. The fraction of initially created NRT-dpa defects that survive is therefore given by the ratio of the outer spherical shell volume to the total cascade volume:

$$\xi_{survive} = \frac{V_{outer} - V_{inner}}{V_{outer}} = \frac{\frac{4\pi R^3}{3} - \frac{4\pi(R-L)^3}{3}}{\frac{4\pi R^3}{3}} = 3\frac{L}{R} - 3\frac{L^2}{R^2} + \frac{L^3}{R^3} \approx 3\frac{L}{R}$$

for $L \ll R$. This “surviving defect production fraction” $\xi_{survive}$ thus tells which fraction of defects predicted by the NRT-dpa model without any recombination survives. Hence we obtain that the damage production taking into account recombination is

$$N'_d(T_d) = \frac{0.8T_d}{2E_d} \xi_{survive} = \frac{0.8T_d}{2E_d} 3\frac{L}{R}$$

The cascade radius R can be, within the regime of spherical cascades, estimated from classical theory of nuclear stopping power^{39,40}. In practise, we used the SRIM code which implements an integral calculation to obtain mean range tables, based on cross sections from the widely accepted Ziegler-Biersack-Littmark (ZBL) interatomic potential⁴⁰

We found that low-energy (less than or of the order to 10 keV) recoils of damage energy T_d has an average movement distance (range) R that is proportional to T_d^x , where the exponent x is $\sim 0.4 - 0.6$ for the metals considered in this study.

Since $\propto T_d^x$, this further gives

$$N'_d(T_d) \propto \frac{0.8T_d}{2E_d} 3\frac{L}{T_d^x} \propto T_d^{1-x}$$

This simple model thus provides an intuitive explanation for why cascade damage production is sublinear with damage energy in the heat spike regime. Physically realistic molecular dynamics simulation studies^{15,29} have reported that defect production rates up to the onset of subcascade formation in a variety of metals can be well described by $N_d \sim (T_d)^m$, where m is between 0.7 and 0.8. These exponent values are slightly larger than the value obtained in our simplified model, likely because real cascades are not perfectly spherical and some defects form small clusters, reducing the recombination probability.

From this derivation and comparison with MD results, one would thus obtain a corrected dpa function (above the threshold) of the form

$$N'_d(T_d) = \frac{0.8T_d}{2E_d} \xi_{survive} = \frac{0.8T_d}{2E_d} AT_d^m$$

where A is a prefactor and $m \approx 0.4 - 0.8$. This form would be valid for spherical cascades. However, it is well known that at high energies cascades break up into subcascades^{25,31}, after which damage production becomes linear with energy. Hence the surviving defect fraction factor $\xi(T_d)$ that accounts also for subcascade breakdown should have the feature of being a power law at low energies, but becoming a constant c at high ones. A function which fulfils both criteria is

$$\xi(T_d) = A'T_d^b + c$$

where $b < 0$ is consistent with the damage production efficiency reducing with increasing energy T_d and the desired limit $\xi(T_d) \rightarrow c$ when $T_d \rightarrow \infty$. This thus gives a total damage production

$$N'_d(T_d) = \frac{0.8T_d}{2E_d} (A'T_d^b + c) = \frac{0.8A'T_d^{1+b}}{2E_d} + \frac{0.8cT_d}{2E_d}$$

Note that here the exponent b is not necessarily the same as m , since the latter ξ function is not a pure power law.

We present here a comparison of the simple power law with the new arc-dpa form (using the constants of Fe as a model case):

As evident from the figure, in the energy range of spherical cascades, < 10 keV, a power law with the exponent of 0.75 is very close to the final arc-dpa functional form. This shows that the arc-dpa power law form is consistent with the derivation and previous MD works.

Very near the threshold (energies < 100 eV) this form is not valid since the concept of a spherical heat spike is not relevant. For backward compatibility we prefer to use the original form NRT-dpa. The prefactor A' is then defined by demanding the function to be continuous, i.e. $\xi(2E_d/0.8) = 1$.

Taken together, this derivation leads us to propose (based in part on review work done within an OECD Nuclear Energy Agency group²²) a modified defect production model, the *athermal recombination corrected displacements per atom* (arc-dpa).

$$N_{d,arcdpa}(T_d) = \begin{cases} 0 & , \quad T_d < E_d \\ 1 & , \quad E_d < T_d < \frac{2E_d}{0.8} \\ \frac{0.8T_d}{2E_d} \xi_{arcdpa}(T_d) & , \quad \frac{2E_d}{0.8} < T_d < \infty \end{cases} \quad (2)$$

with the new efficiency function $\xi_{arcdpa}(T_d)$ given by

$$\xi_{arcdpa}(T_d) = \frac{1 - c_{arcdpa}}{(2E_d/0.8)^{b_{arcdpa}}} T_d^{b_{arcdpa}} + c_{arcdpa} \quad (3)$$

Here E_d is the average threshold displacement energy²⁷ which is the same as in the NRT-dpa and b_{arcdpa} and c_{arcdpa} are fitting constants, that need to be determined for a given material from MD simulations or experiments. The overall form (Eq. 2) and the constant 0.8 are retained for direct comparison with the NRT-dpa model; in particular making it easy to modify computer codes that now use the NRT-dpa by simply multiplying with the function $\xi_{arcdpa}(T_d)$.

In no particular form, the deficiencies of this article are:

R3.2. 1) The title contains acronyms which make no sense to the reader without first reading the manuscript

A3.2. This is a good observation. We now removed the abbreviations and changed the title to make it clear the paper deals with both displacements and replacements (changes marked in red),

R3.3. 2) Figure 1 serves no purpose with regards the new models, and simply presents what is well known in the field. Anyone with an interest in revised NRT models would be well aware of this.

A3.3 While it is true that real experts in the field are aware of this and have seen similar pictures before, the aim of our work is to reach a much broader audience, of people dealing with radiation damage at any level. They may neither be aware of the recombination, or the (much less studied) replacement effect that can enhance precipitate dissolution. For this wider audience, we prefer to keep the figure.

R3.4. 3) Figure 2 is confusing and poorly thought out; Figure 2c should be grouped with Figure 3c & 3d. Figure 2b has lots of data (MD+ experiment) but the legend is a mishmash of acronyms. It present the NRT model data, but not anything from the new model. Figure 2a is a separate concept. The correct conceptually order is i) Figure 2b (explains the problem) , (ii) Figure 2a (i.e. schematic of the new model), iii) Figure 2c + Figure 3.

A3.4. We agree with the referee that the order could be rearranged to be more logical. We have now done this, see new Figs. 2 and 3. We also added a schematic describing the new model (see response A3.1).

R3.5. 4) The explanations of the physical basis of the revised form is incredibly hard to read. Even someone well-versed in the field will struggle to wade through the various scaling arguments. Even if the the description were to be substantially improved, the basic functional form for arcdpa does not seem to be correct. As T_d increases, the functional form should reduce to the linear form as noted at the bottom of page 3. However, this limiting case is not built into the functional form. For

this reason alone the paper should be rejected, as the argument for a physical basis is thrown out the window at the curve fitting stage. The parameter fitting is simply an exercise in fitting data with any old function that works. There is nothing wrong with blind fitting of course, but that is not what is being claimed here. Inspection of the arc-dpa quantities in Table 1 shows that none of the scaling arguments survive into the actual functional form.

A3.5. See response A3.1. above, where we address this concern in detail. We also modified the rpa description slightly, to be better consistent with the arc-dpa description. In particular, now both derivations use the same exponent x , with a clear physical motivation.

R3.6 4a) Note that section 3 does not have the same problems as noted in point (4). Here the functional form has the correct trend for large T_d and the fitted constants have a physically meaningful interpretation.

A3.6. See responses A3.1 and A3.5.

R3.7. 5) There is no mention that the E_d values themselves are often open to interpretation. Certain values have become commonly used, but given that NRT overestimates damage by roughly constant factors, the obvious work-around is to use a higher E_d , or alternatively, use a factor other than 0.8 in Equation 1. It is well-known that the 0.8 is basically a "fudge-factor" which recognizes that the Kinchin-Pease model predicts too many defects.

A3.7. We are well aware of the nature of the threshold displacement energy, as we have published several extensive studies of it. We do not agree, however, that one can use E_d as a fitting factor: it does have a clear physical interpretation, as the average recoil energy needed to induce the formation of one defect. This quantity can and has been measured in several electron beam experiments. For further, very extensive discussions, see e.g. Ref. [Nordlund et al, Nucl. Instr. Meth. Phys. Res. B **246** 322 (2005). (already cited in the manuscript). Although simplified, the factor 0.8 is not fully a fudge factor, but was motivated by binary collision approximation simulations by Robinson et al (reference is in paper).

R3.8. To summarise, the authors are right to attempt to improve on the NRT model, but the present manuscript is not a major enough advance to warrant publication. The heart of the manuscript is essentially an exercise in curve fitting, which could be equally achieved using cubic splines or similar functions. There is no attempt to extract further information based on the parameters, and in the case of Equation 3 the argument developed in the text does not translate to the proposed functional form.

A3.8. We strongly object to this statement. As described earlier in this response, the new constants introduced are not fitting factor, but material constants (see response A2.2) with a physical meaning. Also, the functional forms have a physical motivation (now clarified in the paper) and reasonable low- and high-energy limits. The suggestion by the referee to use "cubic splines" is senseless, since a cubic spline would lead to a limit of plus or minus infinity with increasing energy.

Reviewers' Comments:

Reviewer #1:

Remarks to the Author:

The authors have revised the manuscript and complied with all the suggestions and changes from my previous report. Therefore, in my view, no further modifications are necessary. I consider this manuscript relevant for publication in Nature Communications since it can be of interest to researchers in different disciplines.

Reviewer #2:

Remarks to the Author:

The manuscript represents rather a minor revision of the initial paper and most of my, as well as other reviewers' comments and suggestions have not been followed/implemented to the satisfactory level, but rather rebutted using not convincing, general arguments. Although the paper is relatively well written and the simple ideas that are put forward are clearly explained, as I stressed previously, the research lacks the significant advancement factor and high quality aspect

Comments:

1) The presented results and model are sound and reasonably well justified, although the scientific content lacks the significant advancement factor and high quality of the research. It for sure would be adequate for a regular journal in the respected field, but does not fit into the aims and scope of Nature Communications.

2) My comments and suggestions were not addressed to a satisfactory and most importantly convincing level, so the revised version does not change my opinion on the manuscript and its relevance for Nature Communications.

3) The presented model sounds more like a fitting function than a model (although somehow justified), and does not represent a general solution to the problem with the NRT-dpa model. A significant shortcoming of this study is that Authors neglect the effect of temperature, which should promote the recombination. The effect of temperature should be clearly discussed. It is hard to believe that temperature is not affecting the defect formation/recombination rate in MD simulations and that the coefficients presented in Table 1 are temperature-independent.

4) The proposed model, or correction to the NRT-model, is based and depends on the molecular dynamics simulations data. The four fitting parameters, now called "material constants" which I found also not quite appropriate, have to be fitted to these data. As shown in Table 1, these coefficients vary a lot between different metals and are far from being unique or easy to be associated with a physical effect in a straightforward way as for instance Ed value.

5) As I wrote previously, if authors want to convince the readers that their model represents significant advancement in the fields, they should test it on a larger set of data (MD, experimental) and on more complex materials, including simulations at different temperatures. Six cases is not "several" as Authors still claim in the section 4. Otherwise, corrections (3) and (4) can be seen just as fitting functions that allow for reconciling prediction of a simple NRT-dpa model with the MD data, but not as a general solution to the problem.

6) Authors benchmark on or rather fit to the classical MD simulations results. It should be pointed out that results of these simulations are not unique and may differ with different computational setup (e.g. interaction potentials)

7) Authors call the model: "a very simple and efficient". On the other hand one could call it "more complex". Adding to just one well defined parameter - Ed value - set of four parameter of rather vague physical meaning may be seen as an improvement but not as a simplification.

Some other minor issues:

-) Page 5: "wrt the NRT"???

-) Page 7, "is proportional the sphere"???

Reviewer #3:

None

Reviewer #4:

Remarks to the Author:

I have been asked to provide an opinion on both the manuscript and the previous referee comments. In my opinion, the previous round of reviews addressed all the reviewer's concerns. I would be happy to see this manuscript published in Nature Communications.

A few minor points:

- a significant part of the literature uses the term "thermal spike," rather than "heat spike."

Perhaps the authors could add a note stating that these two terms are interchangeable?

- page 5, sentence beginning with "Recognizing that at low energies...": this is not a complete sentence (verb missing). Please correct.

REVIEWERS' COMMENTS:

Reviewer #1 (Remarks to the Author):

The authors have revised the manuscript and complied with all the suggestions and changes from my previous report. Therefore, in my view, no further modifications are necessary. I consider this manuscript relevant for publication in Nature Communications since it can be of interest to researchers in different disciplines.

We thank the referee for this very favourable judgment.

Reviewer #2 (Remarks to the Author):

The manuscript represents rather a minor revision of the initial paper and most of my, as well other reviewers comments and suggestions have not been followed/implemented to the satisfactory level, but rather rebutted using not convincing, general arguments. Although the paper is relatively well written and the simple ideas that are put forward are clearly explained, as I stressed previously, the research lacks the significant advancement factor and high quality aspect

Comments:

1) The presented results and model are sound and reasonably well justified, although the scientific content lacks the significant advance factor and high quality of the research. It for sure would be adequate for a regular journal in the respected field, but does not into the aims and scope of Nature Communications.

2) My comments and suggestions were not addressed to a satisfactory and most importantly convincing level, so the revised version do not change my opinion on the manuscript and its relevance for Nature Communications.

3) The presented model sounds more like a fitting function than a model (although somehow justified), and does not represents a general solution to the problem with the NRT-dpa model.

We emphasize that we do now present a physically motivated mathematical derivation of the terms in both the arc-dpa and rpa model. Of course this (like any other approximate model in physics) is not a unique “general solution”, but could later on be extended for instance to not be a sharp step function at low energies. We chose not to do this at this stage, because the MD data for low energies (10's of eV's) depends on the interatomic potential, and hence it is not possible to uniquely derive and test a functional form at these energies (by contrast, at the most important keV energies, the MD potentials give consistent trends, which allowed formulating a well-motivated form). We now added the following sentence to the manuscript to address this issue:

We note that when additional and more accurate MD or experimental data becomes available, the models (eqs. (6) and (9)) could be refined for a better description e.g. near the threshold.

A significant shortcoming of this studies is that Authors neglect the effect of temperature, which should promote the recombination. The effect of temperature should be clearly discussed. It is hard to believe that temperature is not affecting the defect formation/recombination rate in MD simulations and that the coefficients presented in Table 1 are temperature-independent.

The issue of temperature was indeed for brevity not discussed extensively in the manuscript. However, there is good reason for this. Many MD studies have addressed the effect of temperature

on the primary damage, and found that as long as the ambient temperature is below about half the melting temperature, the effects are weak. A simple explanation for this is that in the heat spike, the material is heated for a few ps to temperatures around 10 000 K [Zhu et al, Phil. Mag. A. 71 (1995) 735; Nordlund et al, Phys. Rev. B 56 (1997) 2421; Averback and Rubia, Solid State Physics 51 (1998) 281]. Since the core temperature is so high, it does not matter very much whether the surrounding lattice is, say, at 300 K or 600 K. The effect of the ambient temperature has been examined in many works, and these consistently show that from 0 K up to nuclear reactor-relevant temperatures (around 600-700 K), the effect of ambient temperature on damage production is indeed weak [Nordlund et al, Phys. Rev. B 56 (1997) 2421; Phythian et al, J. Nucl. Mater. 223 (1995) 245; Stoller, Comprehensive Nuclear Materials 1 (2012) 293]. Similarly, the ion beam mixing has been experimentally found to be same from 10 to 300 K in almost all materials studies (there are very few mixing studies above 300 K) [Paine and Averback, Nucl. Instr. Meth. Phys. Res. B 7/8 (1985) 666]. Since the dependence is weak, it is clear the arc-dpa or rpa models would give almost the same results for different temperatures.

Even though the effect is weak, it is reasonable to discuss this in the paper. We have now added two data points at 800 K for W into figure 1, and added the following paragraph discussing this into the paper (section 3 Discussion):

We also considered the dependence of the results on the ambient temperature. Several previous studies have shown that the effect of ambient temperature on primary damage production or atom replacements at ps time scales is insignificant or weak up to temperatures around roughly half the melting point^{14,43,44}. For this work, we also simulated two of the data points for W at an elevated temperature, 800 K. The results (solid circles in Fig. 3 b) show that both the damage and replacements is the same within the statistical uncertainty as those at low temperature for the same potential. We note that given sufficiently large and statistically accurate data sets for a range of higher elevated temperatures, it would be possible to make the arc-dpa and rpa model parameters temperature-dependent.

4) The proposed model, or correction to the NRT-model, is based and depends on the molecular dynamics simulations data. The four fitting parameters, now called "material constants" which I found also not quite appropriate, have to be fitted to these data. As shown in Table 1, these coefficients vary a lot between different metals and are far from being unique or easy to be associated with a physical effect in a straightforward way as for instance Ed value.

We do not agree that the parameters would not have a physical meaning. In the modified paper, we already noted that the b_{rpa} has a physical interpretation, the subcascade formation energy.. The derivation presented in the modified manuscript also shows that the exponents $b_{\text{arc-dpa}}$ and c_{rpa} are associated with the dependence of the ion range with energy (which is even an experimentally measurable quantity). Finally, the $c_{\text{arc-dpa}}$ is associated with the saturation value of damage recombination with heat spike size. We admit this was maybe not very clear in the derivation. Hence we now added a summary of this in the beginning of the discussion section:

We first reiterate the physical meaning of the newly introduced material constants. b_{rpa} is related to the subcascade formation energy, and has energy units. The unitless exponents $b_{\text{arc-dpa}}$ and c_{rpa} are associated with the dependence of the ion range with energy. Finally, the unitless quantity $c_{\text{arc-dpa}}$ is associated with the saturation value of damage recombination with heat spike size.

5) As I wrote previously, if authors want to convince the readers that their model represents significant advancement in the fields, they should test it on a larger set of data (MD, experimental) and on more complex materials, including simulations at different temperatures. Six cases is not "several" as Authors still claim in the section 4. Otherwise, corrections (3) and (4) can be seen just as a fitting functions that allow for reconciling prediction of a simple

NRT-dpa model with the MD data, but not as a general solution to the problem.

While it would of course be beneficial to have a large data set for very many different materials, the key point of the paper is to develop a new model of general validity. The good fits shown in the paper to six quite different kinds of metals already give strong evidence of the wide usefulness of the models. Moreover, we note that the original NRT-dpa model, widely used over several decades, was originally developed based on data for only 4 elements [Norgett et al, Nucl. Engr. and Design 33 (1975) 50]

6) Authors benchmark on or rather fit to the classical MD simulations results. It should be pointed out that results of these simulations are not unique and may differ with different computational setup (e.g. interaction potentials)

We do not quite understand why the referee makes this statement, as we already do address exactly this issue explicitly in the manuscript. The data presented in the paper (Figs. 2 and 3) have data sets for 4-5 different interatomic potentials. We also already note explicitly in the text that the results differ due to “due to differences in interatomic potentials”.

7) Authors call the model: "a very simple and efficient". On the other hand one could call it "more complex". Adding to just one well defined parameter - Ed value - set of four parameter of rather vague physical meaning may be seen as an improvement but not as a simplification.

The comparison here is against MD simulations, which take millions of CPU hours to run and require significant supercomputer capacity. Being able to use a single analytical equation with three parameters is certainly a huge simplification compared to such a simulation effort.

Some other minor issues:

-) Page 5: "wrt the NRT"???

This abbreviation has now been spelled out as **with respect to**.

-) Page 7, "is proportional the sphere"???

We added **to** after proportional.

Reviewer #4 (Remarks to the Author):

I have been asked to provide an opinion on both the manuscript and the previous referee comments. In my opinion, the previous round of reviews addressed all the reviewer's concerns. I would be happy to see this manuscript published in Nature Communications.

We thank the referee for this very favourable judgment.

A few minor points:

- a significant part of the literature uses the term "thermal spike," rather than "heat spike." Perhaps the authors could add a note stating that these two terms are interchangeable?

This is a good point. We now added “**(also known in parts of the literature as ‘thermal spike’)**” the first time the term is mentioned.

- page 5, sentence beginning with "Recognizing that at low energies...": this is not a complete sentence (verb missing). Please correct.

Indeed this sentence was unduly complicated and probably grammatically wrong. We simplified it to read “**At low energies** (below the subcascade formation regime14) (Sto12) the displacement cascades are roughly spherical with radius R, **and form** a liquid-like zone of dense collisions (the heat spike described above).”

*** See Nature Research's author and referees' website at www.nature.com/authors for information about policies, services and author benefits*

This email has been sent through the Springer Nature Tracking System NY-610A-NPG&MTS

Confidentiality Statement:

This e-mail is confidential and subject to copyright. Any unauthorised use or disclosure of its contents is prohibited. If you have received this email in error please notify our Manuscript Tracking System Helpdesk team at <http://platformsupport.nature.com> .

Details of the confidentiality and pre-publicity policy may be found here <http://www.nature.com/authors/policies/confidentiality.html>

Privacy Policy | Update Profile <